# Molecular understanding of the critical role of alkali metal cations in initiating CO₂ electroreduction on Cu(100) surface

Zhichao Zhang ®[1], Hengyu Li[1], Yangfan Shao[1], Lin Gan ®[1], Feiyu Kang[1] ✉,
Wenhui Duan[2,3,4], Heine Anton Hansen ®[5] & Jia Li ®[1] ✉

Molecular understanding of the solid–liquid interface is challenging but essential to elucidate the role of the environment on the kinetics of electrochemical reactions. Alkali metal cations (M⁺), as a vital component at the interface, are found to be necessary for the initiation of carbon dioxide reduction reaction (CO₂RR) on coinage metals, and the activity and selectivity of CO₂RR could be further enhanced with the cation changing from Li⁺ to Cs⁺, while the underlying mechanisms are not well understood. Herein, using ab initio molecular dynamics simulations with explicit solvation and enhanced sampling methods, we systematically investigate the role of M⁺ in CO₂RR on Cu surface. A monotonically decreasing CO₂ activation barrier is obtained from Li⁺ to Cs⁺, which is attributed to the different coordination abilities of M⁺ with *CO₂. Furthermore, we show that the competing hydrogen evolution reaction must be considered simultaneously to understand the crucial role of alkali metal cations in CO₂RR on Cu surfaces, where H⁺ is repelled from the interface and constrained by M⁺. Our results provide significant insights into the design of electrochemical environments and highlight the importance of explicitly including the solvation and competing reactions in theoretical simulations of CO₂RR.

The electrochemical carbon dioxide reduction reaction (CO₂RR) has emerged as an attractive method for simultaneously closing the carbon cycle and producing important industrial feedstocks due to its compatibility with renewable, carbon-free energy sources[1,2]. Cu-based materials are particularly unique among the various types of CO₂RR catalysts developed today, as they can produce multi-electron (>2 e⁻) reduced products like methane[3], as well as multi-carbon (C₂₊) products consisting mainly of ethylene[4,5] and ethanol[6], regardless of their different structures, such as Cu foil[7], nanoparticles[8], and even molecular catalysts[9]. However, the activity and selectivity of these catalysts have not yet reached the level required for industrial application. Therefore,

many efforts have been made to improve their electrochemical performance further, especially for more valuable C₂₊ products. These efforts include facet design[10], defect engineering[11], pretreatments such as oxide-derived Cu (OD-Cu)[12], and electrolyte engineering[13,14]. However, the determining factors and atomic-level insights into the underlying mechanisms are still unclear.

CO₂RR, as a representative aqueous electrochemical reaction, is significantly affected not only by the catalyst but also by the environment at the solid–electrolyte interface. Since the 1990s, Hori et al. have discovered that alkali metal cations (M⁺) can enhance CO₂RR activity by more than an order of magnitude and concurrently improve C₂₊/C₁

[1]Shenzhen Geim Graphene Center and Institute of Materials Research, Tsinghua Shenzhen International Graduate School, Tsinghua University, Shenzhen 518055, People's Republic of China. [2]State Key Laboratory of Low Dimensional Quantum Physics and Department of Physics, Tsinghua University, Beijing 100084, People's Republic of China. [3]Institute for Advanced Study, Tsinghua University, Beijing 100084, People's Republic of China. [4]Frontier Science Center for Quantum Information, Beijing 100084, People's Republic of China. [5]Department of Energy Conversion and Storage, Technical University of Denmark, Kgs, Lyngby 2800, Denmark. ✉e-mail: fykang@sz.tsinghua.edu.cn; li.jia@sz.tsinghua.edu.cn

selectivity on the Cu electrode. This promoting effect becomes more significant with larger cations from Li+ to Cs+[13]. The promoting effect of alkali metal cations has been confirmed[15–17] and extended to other CO2RR catalysts, such as CO-selective Ag[18], Au[19], and HCOOH-selective SnO2[20]. Several explanations have been developed so far. The surface pH model attributed the differences between alkali metal cations to the interfacial proton and CO2 concentrations resulting from the equilibration between carbonate-group anions[13,18], but the relationship between this model and the reaction mechanism is not well understood. Subsequently, a dipole-field interaction model was proposed, which emphasizes the stabilizing effect of the electric field between solvated alkali metal cations and the negatively charged metal surface on polar intermediates[15,21,22]. Therefore, intermediates with large dipoles involving *CO2 and *OCCO (the asterisk * denotes the adsorption site) would be strongly stabilized, enhancing the selectivity of corresponding products such as CO on Ag/Au and C2+ on Cu. However, this model is based on the uniform interfacial electric field model and attributes the discrepancy between different alkali metal cations to the tendency of larger cations to accumulate more at the interface while ignoring the intrinsic difference of individual alkali metal cations and possible local interactions between cations and intermediates.

Recently, it has been further demonstrated that the CO2RR cannot be initiated on Cu, Ag, Au, and SnO2 without alkali metal cations, even to CO and HCOOH[20,23,24], which cannot be fully explained by the above models. In addition, spectroscopic results have shown that the introduction of Li+, but not Cs+ cations, results in the highest electric field in the Stern layer, while the strength of the solvation-induced reaction field, the Onsager field, varies with the cation size[25]. These findings suggest that different alkali metal cations have inherent differences rather than a simple concentration disparity and that cations at the interface could greatly influence the efficiency of the CO2-to-CO conversion process, potentially playing a more active role rather than merely acting as spectators. To provide a more detailed explanation of the influence of solvated alkali metal cations on reaction mechanisms, ab initio molecular dynamics (AIMD) simulations were performed and revealed the existence of local interactions, such as coordination between solvated cations and surface intermediates[26]. Notable examples include the interaction between K+ cation and adsorbed *CO2 molecule on Au surface[23,27], and the interaction between Li+/K+/Cs+ cations and *CO + *CO dimer on Cu surface[28]. However, a comprehensive understanding of the effect of different alkali metal cations on

CO2 activation and their critical role in CO2RR initialization on the Cu surface is still lacking.

In this work, we have systematically studied the effect of alkali metal cations on both CO2RR and the main competing hydrogen evolution reaction (HER), using AIMD simulations with enhanced sampling methods and explicitly taking into account the effects of solvation and potential. By calculating the free energy barriers of CO2 activation, a monotonically decreasing trend from Li+ to Cs+ is found, which is due to the coordination ability between the alkali metal cation and the *CO2 intermediate. The bridge coordination between larger cations and *CO2 helps to facilitate CO2RR more compared to the side configuration preferred by smaller cations. Furthermore, we show that the necessary role of alkali metal cations in CO2RR on Cu surfaces cannot be fully understood without simultaneously considering the competing HER. Our results shed light on the intrinsic short-range stabilizing effect of individual alkali metal cations on CO2RR and highlight the necessity of explicitly considering the environment, such as cation and solvation, as well as competing reactions, in electrochemical simulations in order to obtain more reliable insights.

## Results and discussion
### Electrical double-layer response of applied potential
To provide a benchmark for the subsequent analysis, we first performed simulations of the Cu–water interface system with and without CO2 adsorption, denoted as Cu–*CO2–H2O and Cu–H2O, respectively, to investigate the response of the electrical double layer (EDL) to different applied potentials. Figure 1a–c shows the density distribution of water perpendicular to the interface under different conditions. Three primary peaks could be observed above the Cu–water interface at approximately 2.1, 2.8, and 5.2 Å, respectively. The first peak (marked with the gray dashed line in Fig. 1a–c) is attributed to water molecules chemisorbed on the Cu surface, while the second and third peaks indicate two partially ordered water layers. No evident peaks were observed further away from the interface, representing the diffusion layer and the bulk water with a density of about 1 g cm−3, and indicating the sufficient thickness of the water layer as well. Furthermore, as the applied potential decreases, the intensity of the first peak decreases, as shown in Fig. 1a, which could be attributed to the increasing repulsive interaction between the more negatively charged substrate and the partially negatively charged O atom of the H2O molecule (as schematically shown in Fig. 1d). Meanwhile, H atoms would be attracted by the

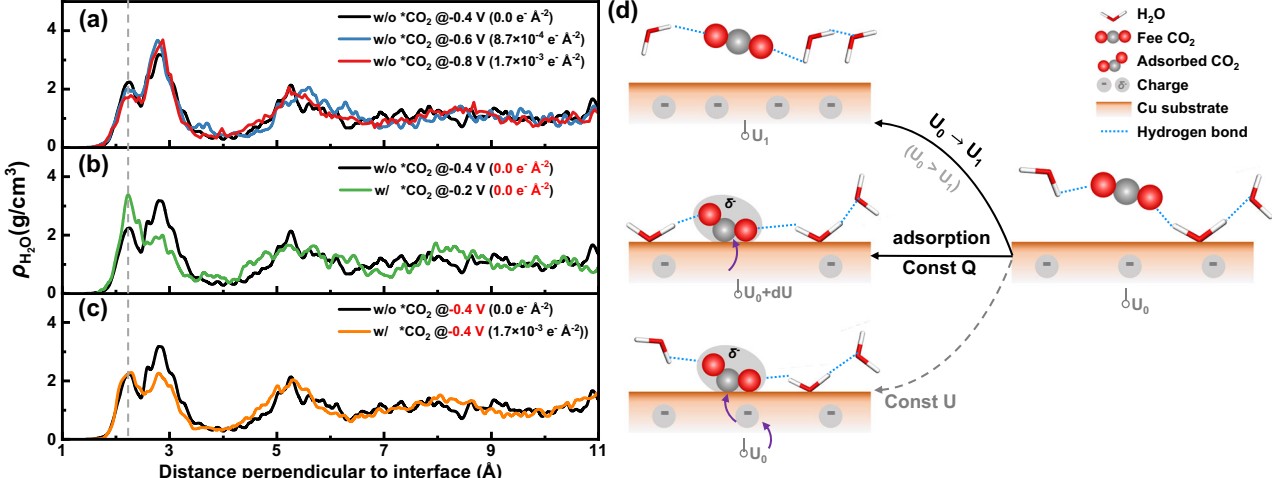

**Fig. 1 | Response of interfacial structure to the applied potential.**
**a**–**c** Distribution of water density along the direction perpendicular to the Cu(100)–water interface under different conditions, with the topmost Cu layer at the distance of 0. The gray dashed line indicates the first peak of chemisorbed water. **d** Schematic of the interfacial structure change under corresponding conditions to (**a**–**c**). CO2 molecules are depicted as gray and red spheres, while H2O molecules are represented as stick models for clarity. The applied potentials are obtained from the work function of the whole system. (Supplementary Note 1).

**Fig. 2 | Schematic of the representative *$CO_2$ adsorption structures.** Local interactions between *$CO_2$ at Cu(100) surface and the surrounding environment: **a** Schematic of the bidentate adsorption of *$CO_2$ at Cu surface in an explicit solvent with C and one of the O atoms coordinating with Cu, and the other O atom forming hydrogen bonds (light blue dashed lines) with surrounding water. **b** Schematic of the side configuration of $Li^+$ or $Na^+$ coordinated with only one O atom of *$CO_2$ (magenta dashed line). **c** Schematic of the bridge configuration of $K^+$ or $Cs^+$ coordinated with both two O atoms of *$CO_2$.

surface, causing the O–H bond to rotate toward the surface, as reflected by the change in interfacial water dipoles within 5 Å of the Cu surface (Supplementary Fig. 1). The dipole distribution changes from random to a single peak, indicating a more aligned distribution of water molecules in the first water layer.

A similar response of the interfacial water arrangement to the applied potential was also observed in the Cu–*$CO_2$–$H_2O$ system (Supplementary Fig. 2), where a ˙$CO_2$ molecule was introduced onto the Cu surface of the Cu–$H_2O$ system. Energetically, the adsorption enthalpies of $CO_2$ decrease with decreasing applied potential, indicating the more stable adsorption of $CO_2$ (Supplementary Table 1). During the additional 15 ps of AIMD simulation, $CO_2$ was observed to be stably adsorbed on the Cu surface in a bidentate configuration through the C–Cu and O–Cu bonds, with the other O atom facing upward and forming hydrogen bonds with the surrounding water molecules (adsorption configuration in Fig. 1d). Bader charge analysis[29] shows that the chemisorbed $CO_2$ has a charge of about −1.05 $e$, indicating a fully activated adsorption state. In addition, the introduction of $CO_2$ enhances the adsorption of chemisorbed water (Fig. 1b), but this is due to the lack of potential control (Const-Q case in Fig. 1d). After adjusting the applied potential to be consistent with the Cu–$H_2O$ system by electron exchange with the electron reservoir, the intensity of the first peak is almost identical to the reference (Fig. 1c and Const-U case in Fig. 1d). The above results confirm the validity of our model and provide the basis for further investigation of the electrochemical interface during the $CO_2$RR process[30–33].

Furthermore, it is worth noting that the bidentate configuration of *$CO_2$ on the Cu(100) surface differs from that observed in a fully implicit solvent environment. In the latter case, $CO_2$ tends to be physisorbed on the Cu(100) surface over a wide potential range (Supplementary Fig. 3), while the bidentate configuration is less stable by ~0.3 eV at −0.6 V vs. SHE (standard hydrogen electrode), which is the concerned potential range in this work. This difference indicates the significant contribution of surrounding hydrogen bonds to the stabilization of *$CO_2$ from explicit solvation and thus raises a cautionary note regarding the use of fully implicit solvents for investigating molecular explanations of reaction mechanisms, especially when the interactions between intermediates and the surrounding electrochemical environment, such as solvated water molecules, play a substantial role.

## Promoting effects of alkali metal cations on $CO_2$ adsorption

Experimental results have shown that $CO_2$RR products, including CO, cannot be detected on coinage metals without $M^+$ in the electrolyte[20,23,24]. This suggests that $CO_2$RR is inhibited in the early stages of the $CO_2$-to-CO conversion process. To investigate the underlying mechanism, here we first focused on the effect of alkali metal cations on the rate-limiting step of $CO_2$RR to CO, the $CO_2$ activation process[34–36], which is expected to have electrostatic interactions

with alkali metal cations due to its transformation from a linear non-polar configuration to a bent polar one. We studied the Cu–*$CO_2$–$H_2O$ system in more detail and subsequently added $M^+$–$F^-$ ($M^+$ = $Li^+$, $Na^+$, $K^+$, or $Cs^+$) ion pairs into it to keep the applied potential close to the PZC of Cu(100) as well as the onset potential of $CO_2$RR at Cu (−0.6 V vs. SHE)[14], which are referred to as Cu–*$CO_2$–$M^+$(aq) systems. Bader charge analysis confirms full ionization of the added ions, with +0.9 $e$ for alkali ions and −0.85 $e$ for F ions, respectively. Using the angle of $CO_2$ as a collective variable (CV) during the adsorption process, we found that the CVs of the initial state (IS, a free linear $CO_2$) and the final state (FS, a chemisorbed bent $CO_2$) do not vary much with different cations, peaking at -174.5° and -118°, respectively (Supplementary Fig. 4). The angle of $CO_2$ at the IS deviates from 180° due to thermal fluctuations and interactions with water, and the chemisorbed configuration of *$CO_2$ at the FS is identical to that in the Cu–*$CO_2$–$H_2O$ system (Fig. 2 and Supplementary Fig. 4). These results further demonstrate the significant role of hydrogen bonds formed between *$CO_2$ and surrounding $H_2O$ molecules.

More importantly, we found that all alkali metal cations can stably coordinate with *$CO_2$ but with different coordination features. $Li^+$ and $Na^+$ ions coordinate with only one O atom of *$CO_2$ (side configuration, as shown in Fig. 2b), while $K^+$ and $Cs^+$ would coordinate with both O atoms of *$CO_2$ (bridge configuration, as shown in Fig. 2c). One could naturally assume that the bridge coordination results from the larger cation radius, which is supported by the increasing first solvation shell radius ($r_{M(H_2O)}$) from $Li^+$ to $Cs^+$ (Supplementary Table 2). If the configuration differences are maintained during the activation process, this phenomenon could be an intrinsic cause of the different promoting effects on $CO_2$RR of $M^+$ with different sizes. The bridge configuration is more favorable for $CO_2$ bending as a consequence of the electrostatic repulsion between $M^+$ and the partially positively charged C atom of $CO_2$. From an energetic point of view, we firstly considered the $CO_2$ adsorption free energy with different alkali metal cations as an approximation of the activity descriptor. The system with $Li^+$ shows the strongest adsorption of *$CO_2$, while the adsorption free energies with the other three cations do not differ much from each other (Fig. 3 and Supplementary Table 3). Consequently, no clear trend with cation size could be observed, which is also consistent with previous results on Au(111)[23]. As repeatedly demonstrated[35,37], the reaction energy alone cannot fully describe the activity of a system, while the activation energy from the IS to the transition state (TS) might be more important to serve as the real energetic criteria. Therefore, in order to obtain more accurate and detailed information about the cation effects on the $CO_2$ adsorption process, we next concentrated on the free energy barrier and transition states of the $CO_2$ adsorption process at Cu(100).

Figure 3a shows the free energy profile of $CO_2$ activation in the Cu–*$CO_2$–$H_2O$ system in the absence of $M^+$, where a kinetic barrier of 0.22 eV is indicated. For comparison, Fig. 3c–f shows the free energy profiles of $CO_2$ activation with ions in the solvent. The kinetic

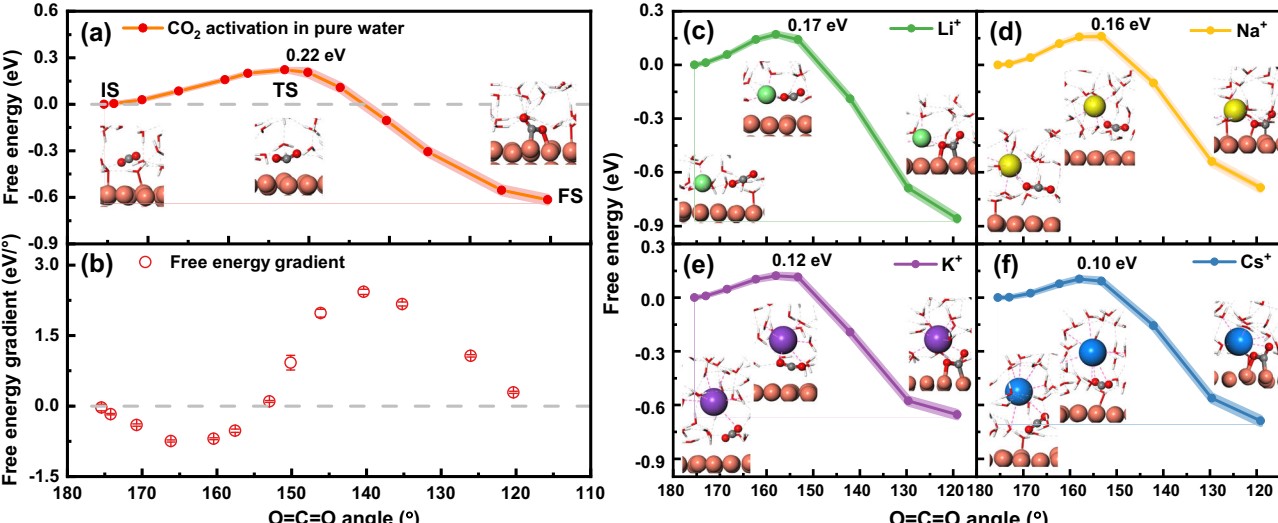

**Fig. 3 | Free energy profiles and representative structures for CO₂ activation process on Cu(100) surface. a** Free energy profile and representative initial state (IS), transition state (TS), and final state (FS) structures for the CO₂ adsorption process in the pure water (Cu–*CO₂–H₂O) system without M⁺. The gray dashed lines indicate the zero value. **b** Corresponding free energy gradient with error bar at each calculated collective variable. Crossover points with dashed baseline indicate the positions of IS, TS, and FS. The error bars represent standard errors of free energy gradients calculated using block average at the corresponding collective variable. **c–f** Free energy profiles and representative structures at IS, TS, and FS for CO₂

adsorption with different cations (Cu–*CO₂–M⁺) of **c** Li⁺, **d** Na⁺, **e** K⁺, and **f** Cs⁺. The shadows in all five free energy profiles represent the cumulative errors ( ~ 0.01 eV at TS) estimated with the block average method (Supplementary Notes 2 and 3). In the local structures, CO₂ molecules are depicted as gray and red spheres, Cu substrates are shown as reddish-brown spheres, H₂O molecules are illustrated as red and white sticks, and M⁺, including Li⁺, Na⁺, K⁺, and Cs⁺ are represented by green, yellow, purple and blue spheres, respectively. The magenta dashed lines around M⁺ illustrate their coordination with O atoms in the first solvation shells.

barriers exhibit a clear, monotonically decreasing trend from the Cu–*CO₂–H₂O system to the Cu–*CO₂–M⁺(aq) systems, with the inclusion of Li⁺, Na⁺, K⁺, and Cs⁺. This trend is consistent with the observed activity trend of CO₂RR with alkali metal cations in experiments[15–19]. Moreover, cations are generally divided into two groups during the activation process based on the activation barriers and their coordination characteristics with *CO₂: smaller cations of Li⁺/Na⁺ and larger cations of K⁺/Cs⁺ (validation of the stable configurations is discussed in Supplementary Note 4). The free energy barrier difference between the two groups is about 0.05 eV, which can increase the corresponding reaction rate by a factor of 5, which is comparable to the experimental results[15–19,25].

The activation barrier in the CO₂ bending process and the concerted charge transfer result from the degeneracy breaking of the 2πᵤ (lowest unoccupied molecular orbital, LUMO) and 1π_g (highest occupied molecular orbital, HOMO) states of CO₂. Concurrently, the lower LUMO level of the bent CO₂ facilitates the back donation of electrons from the substrate to CO₂, resulting in the negatively charged *CO₂[−36]. This process can be further enhanced by alkali metal cations (Fig. 3). In the Cu–*CO₂–H₂O system, the CV at the TS is approximately 153°, but it moves closer to the IS with alkali metal cations. Meanwhile, the negative charges on the CO₂ at the TS show a similar monotonically decreasing trend, with about −0.3 *e* transferred to CO₂ in the Cu–*CO₂–H₂O system, while only about −0.17 *e* is transferred in the Cu–*CO₂–Cs⁺(aq) system (Supplementary Table 4). This also implies that the TS is closest to the IS with the Cs⁺ cation, which benefits its lowest activation barrier.

The coordination structures of M⁺ in the first solvation shell with and without *CO₂ at the TS were further analyzed to understand the underlying molecular mechanism of the CO₂ activation (Fig. 4a), with the systems denoted as Cu–*CO₂–M⁺(aq) and Cu–M⁺(aq), respectively. For Li⁺ and Na⁺ cations, the coordination with CO₂ involves the exchange of an oxygen atom from H₂O in the first solvation shell, resulting in an apparently unchanged total coordination number. This exchange is mainly due to the hard solvation shells of small cations,

which can hardly accept more solvation molecules with the environment, as shown by the platform in the integrated radial distribution function (RDF) at the radius of the first solvation shell in Supplementary Fig. 5. However, for K⁺ and Cs⁺ cations, the coordination numbers increased by 0.9 and 2.2, respectively. This is because K⁺ and Cs⁺ could coordinate with more than one oxygen from *CO₂ and maintain the coordination with H₂O. The higher coordination number of K⁺ and Cs⁺ cations with their unique bidentate coordination structure with *CO₂ could facilitate the electron transfer process and help to activate CO₂ more effectively than smaller cations, providing important insights into the molecular origin of their higher promoting effect on CO₂RR activity. In addition, there is a difference in the CO₂ activation barrier between Cs⁺ and K⁺ cations, but experimentally, they were found to activate CO₂ similarly[25], which could also be explained by their coordination structure. Comparing the total coordination numbers of alkali metal cations in the Cu–M⁺(aq) system with those in the Cu–*CO₂–M⁺(aq) systems (Supplementary Fig. 6), most remain unchanged except for a decrease in the Cs⁺ cation at the interface without *CO₂. In the AIMD simulation, we observed that the Cs⁺ cation loses some solvated H₂O molecules at the Cu–M⁺(aq) interface (Supplementary Fig. 7), but when it coordinates with *CO₂, its solvation ability with H₂O is slightly recovered at the Cu–*CO₂–M⁺(aq) interface (Fig. 4a). The partially desolvated Cs(H₂O)ₓ complex behaves like quasi-specific adsorption that may block the active sites and be unfavorable for surface reactions, ultimately resulting in its similar activating effect on CO₂ with K⁺.

Figure 4b further shows the statistical distribution of the distance between alkali metal cations and two oxygens of CO₂, O₁, and O₂. Li⁺ and Na⁺ cations were observed to coordinate stably with only one oxygen, resulting in a larger M⁺–O₂ distance compared to M⁺–O₁. In contrast, for K⁺ and Cs⁺ cations, the preferred bridge configuration leads to an equal distance of M⁺–O₁ and M⁺–O₂. Another difference between these two cation groups is the density of the distributions, with the larger cations showing a broader distribution, consistent with the absence of platforms in the integrated RDF (Supplementary Fig. 5c,

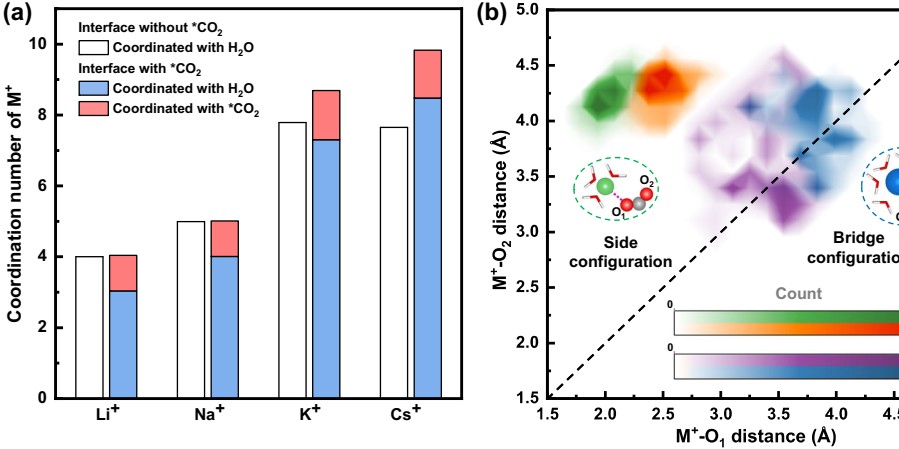

**Fig. 4 | Coordination analysis of the first solvation shell of alkali metal cations around TS. a** The number of coordinated O for $Li^+$, $Na^+$, $K^+$, and $Cs^+$ cations at the $Cu(100)$−electrolyte interface with and without the $*CO_2$. **b** Contour map of the distance distribution between alkali metal cations and two oxygens of $CO_2$, statistically derived from snapshots around the transition state. The results of $Li^+$, $Na^+$, $K^+$, and $Cs^+$ are represented by green, orange, purple, and blue, respectively.

d). This suggests a more dynamic coordination around $CO_2$ compared to $Li^+$ and $Na^+$ cations. In addition, we observed an accumulation of electrons between the $Li^+$ cation and the coordinated O of $CO_2$, which is the characteristic feature of a bond. However, no significant charge accumulation was observed for the $Cs^+$ cation (Supplementary Fig. 8), supporting the above analysis. In this way, more dynamically coordinated $K^+/Cs^+$ with $*CO_2$ could also accelerate the reaction cycle by flexibly moving away to help activate $CO_2$ at other active sites and avoid inhibiting further reaction steps compared to more strongly coordinated $Li^+/Na^+$.

Based on the kinetic barrier simulations of $CO_2$ activation and the coordination analysis of $M^+$, we could conclude that the distinct coordination ability of alkali metal cations is responsible for their different promotional roles in $CO_2$ activation on the Cu surface. The bridge configuration between $K^+/Cs^+$ and $*CO_2$ could both facilitate the $CO_2$ bending and electron transfer process, and the flexible coordination between them could accelerate the reaction cycle. These effects together result in the obvious activity and selectivity enhancement of $K^+$ and $Cs^+$ cations compared with $Li^+$ and $Na^+$. These results are consistent with the trend of Onsager reaction field strengths from the spectroscopic exploration[25].

Furthermore, similar behavior could also be expected on other coinage metal surfaces or facets due to their similar response to alkali metal cations[18,19]. Basically, the effect of different metal surfaces or facets on an electrochemical system can be divided into two main aspects. First, it can alter the active sites, thereby affecting the binding strengths between the substrate and intermediates due to changes in substrate properties or the coordination number of the active sites. Second, it can alter the potential of the zero charge, thus influencing the surface charge density under specific applied potentials. Since the activation of $CO_2$ involves electron transfer processes, changes in the surface charge density can significantly affect the driving forces provided by the applied potentials. For example, in our simulations, we observed stable adsorption of bent $*CO_2$ on $Cu(100)$ surfaces without any constraints or external factors such as cations or potentials. In contrast, previous reports on the $Au(111)$ surface indicated that bent $*CO_2$ could not be stably adsorbed even in the presence of $K^+$ ions[23,37]. However, it is further demonstrated that bent $*CO_2$ can become a stable configuration at higher surface charge densities on the $Au(110)$ surface[27]. Despite the different stable adsorption configurations observed on different metal surfaces, the promoting role of alkali metal cations remains a consistent factor. Therefore, we expect that some of the findings from our work can be applied to other metal surfaces and facets and that additional effects will also be explored in our future work.

## Necessary effect of $M^+$ on CO2RR

The above analysis successfully explains the increasingly promoting role of alkali metal cations from $Li^+$ to $Cs^+$ on $CO_2$ activation. However, the relatively low kinetic barrier of 0.22 eV in the $Cu-*CO_2-H_2O$ system cannot fully account for the necessary role of alkali metal cations on CO2RR observed in the experiment. To determine this, we need to consider not only the main reaction $CO_2$ activation but also the side reaction HER. In the absence of alkali metal cations, the highly acidic electrolytes used in the experiment could lead to a high concentration of surface protons, causing the competing HER to dominate[20,23,24]. Although HER is known to be a major competing side reaction during CO2RR, the molecular understanding of the effects of alkali metal cations on HER has rarely been studied. Therefore, we further investigated the possible influence of alkali metal cations on proton accessibility and HER activity on the Cu surface.

Figure 5 shows the distributions of an excess proton with and without the alkali metal cations at the interface. Cumulative averages of the results are shown in Supplementary Fig. 9. The excess proton is solvated to form a hydronium ion ($H_3O^+$), which can move along the hydrogen bond network by hopping between different water molecules. After equilibrating for more than 20 ps, the solvated proton reaches a dynamically stable configuration at a distance of about 5.2 Å away from the Cu surface. Based on the water distribution shown in Fig. 1, the excess proton diffuses into the second water layer (third peak in Fig. 1a), which satisfies the three hydrogen bonds required by the $H_3O^+$ ion. However, when alkali metal cations are introduced at the interface, the proton-interface distance increases, except for $Li^+$ (Fig. 5a). The change in proton-interface distance results from the disruption of the interfacial hydrogen bond network and the electrostatic repulsion of solvated alkali metal cations[24,38,39]. For larger cations, including $Na^+$, $K^+$, and $Cs^+$, the excess proton is repelled away from the second water layer and confined in the first solvation shells of these cations, as indicated by the comparable proton-interface distances and the diameters of the first solvation shells of alkali metal cations. The average proton-interface distance with $Cs^+$ is slightly less than twice the radius of the first solvation shell due to its partial desolvation behavior, similar to the case in Supplementary Fig. 7. As for the case with $Li^+$, the proton-interface distance seems to be even smaller than in the case with a single $H_3O^+$, which is due to two aspects. On the one hand, the solvation shell of $Li^+$ is smaller and harder than that of larger alkali metal cations, as indicated in the previous sections, which means that the interfacial hydrogen bond network is relatively more complete, leaving space for $H_3O^+$ to form three required hydrogen bonds at the interface. In this way, the $H_3O^+$ and solvated $Li^+$ could coexist

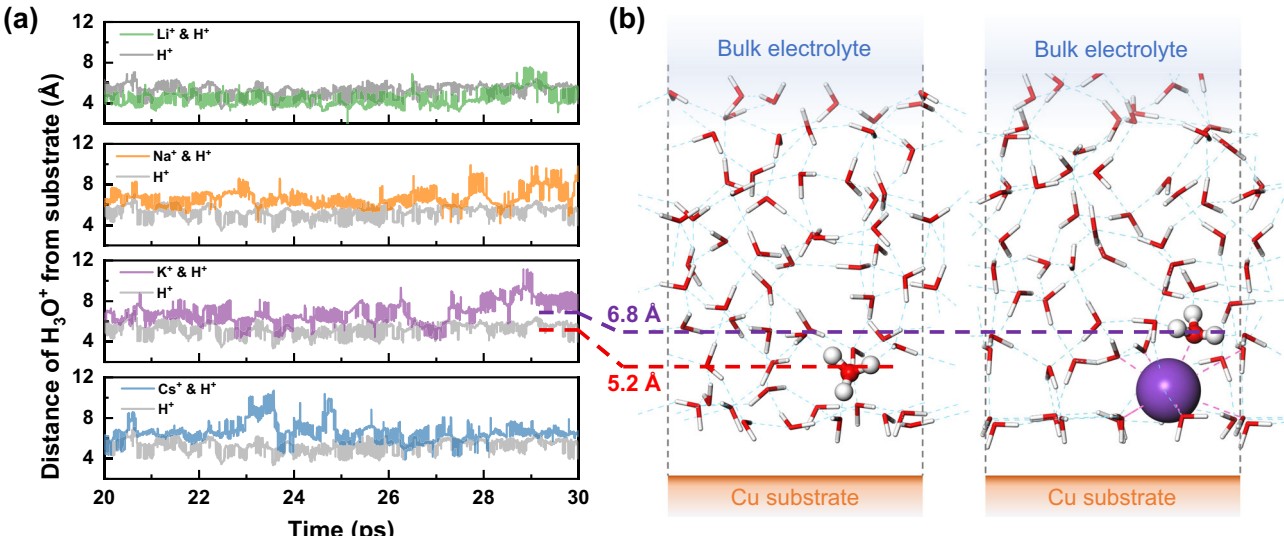

**Fig. 5 | Interaction between solvated proton ($H_3O^+$) and alkali metal cations.** **a** Distribution of $H_3O^+$ in the solvent with and without alkali metal cations. The red and purple dashed lines represent the average distance of the excess proton from the Cu surface in the production timescale of 10 ps with and without $K^+$ cation, respectively. The corresponding representative interface structures are shown in **(b)**. $H_3O^+$ and $K^+$ are highlighted and shown as spheres where O is represented in red, H in white, and K in purple, while other atoms are hidden for visual clarity. Magenta dashed lines around $K^+$ illustrate the coordination with O atoms in the first solvation shell.

between the first and second water layers despite of the electrostatic repulsion between them. Furthermore, after the introduction of alkali metal cations, the negative charge density at the Cu(100) surface increases compared to that of single $H_3O^+$, which further attracts the proton to approach the interface. Therefore, the attraction from the negatively charged catalyst surface and the inhibition from alkali metal cations combine to result in the distinct proton-interface distances with various alkali metal cations near the interface. Furthermore, the hydrogen bond lifetime decreases with the inclusion of the $Li^+$ cation and is further suppressed with increasing cation size (Supplementary Note 5), which provides additional support for the influence of solvated alkali metal cations on solvent dynamics.

In addition, since $H_2O$ molecules could also be potential proton sources, we further investigated the kinetic barriers of the Volmer step from $H_2O$ molecules, such as free $H_2O$, surface-adsorbed $H_2O$, and $K^+$-coordinated $H_2O$, as shown in Supplementary Fig. 10. The free and adsorbed $H_2O$ molecules exhibit high free energy barriers of 0.75 eV and 0.95 eV, respectively, while the $K^+$-coordinated $H_2O$ shows the highest barrier of 1.05 eV, which are all far more difficult to happen than $CO_2RR$. Therefore, by combining the analysis of cation effects on both $CO_2RR$ and competing HER reactions, we were able to fully explain the necessary role of alkali metal cations in $CO_2RR$, which resulted from the inhibition of protons derived from $H_3O^+$ approaching the surface, as well as the much higher kinetic barrier of protons derived from $H_2O$ molecules compared to $CO_2$ activation.

Note that only the qualitative trend is provided here. Typically, determining the reaction network requires a quantitative criterion based on differences in activation barriers. However, in this case, the critical factor is the inhibition of protons approaching the surface by cations. This implies that the concentration of reactants, influenced by potential, ionic activities, and mass transport, may play the dominant role. Furthermore, in reality, the interfacial concentration of alkali metal cations may deviate from one another under a specific potential due to different capacitance responses[40–43]. Since only a few ions can be introduced at the nanoscale simulations due to scale limitations, it is challenging to reflect the concentration change accurately. Hence, this necessitates further exploration of both interfacial evolution and simulation methods.

Furthermore, our results derived from the fully explicitly solvated interfacial model offer significant insights into future electrochemical simulation work. Notably, the model indicates different adsorption configurations of the $CO_2$ molecule in implicit and explicit solvents, distinct coordination structures of solvated alkali metal cations and intermediates, and the interaction between protons and alkali metal cations. The results suggest the importance of explicitly considering the electrolyte in simulation models, particularly when dealing with species that could potentially interact with the interfacial hydrogen bond network or other components within the electrical double layer, including solvated ions. Meanwhile, our investigations into the copper surface may aid in gaining a broader understanding of how alkali metal cations impact $CO_2RR$ performance. This research can be combined with studies conducted on gold surfaces[27,37,44], where the alkali metal cations could differentiate the $CO_2$ activation pathways between the inner- and outer-sphere mechanisms. After clarifying the various roles that alkali metal cations play on different coinage metals and even other $CO_2RR$ catalysts, it is expected that a more systematic relationship between EDL design and $CO_2RR$ performance can be established, which demands further exploration.

In conclusion, the results of a series of explicitly solvated AIMD simulations on $CO_2RR$ and competing HER have provided systematic insights into the promotional roles and the necessary effects of alkali metal cations for $CO_2RR$. Energetically, the $CO_2$ activation barrier decreases with increasing cation size. The different promotional roles can be attributed to the coordination abilities, or Onsager field strengths, of various alkali metal cations. In particular, smaller $Li^+$ and $Na^+$ cations can only coordinate with one oxygen atom of $CO_2$ and simultaneously release a water molecule, whereas $K^+$ and $Cs^+$ cations can flexibly coordinate with both oxygen atoms of $CO_2$, which is more favorable for $CO_2$ activation and reaction cycling. However, the partial desolvation of $Cs^+$ cation slightly inhibits the increasing tendency, resulting in its similarity with $K^+$ cation. On the other hand, the necessary role is attributed to the competition between HER and $CO_2RR$. The interfacial proton accessibility can be largely hindered by alkali metal cations, and the much higher kinetic barrier of protons derived from $H_2O$ molecules helps $CO_2RR$ to outcompete HER. Given the similarity in catalytic behavior with alkali metal cations, these phenomena are expected for other CO-selective catalysts, such as Ag

and Au. Our results help to understand the molecular origin of the possible role of the electrochemical environment emphasize the importance of explicit solvation, ions, and systematic exploration of competing reactions in electrochemical simulations, and show the effectiveness of theoretical tools in exploring the solid–electrolyte interface.

## Methods

### Solid–electrolyte–Ne double electrode model

Based on experimental results, it was determined that the Cu(100) surface would dominate after a prolonged electroreduction process[45]. To represent the electrode, we chose the Cu(100) surface modeled as a three-layer slab with the p(3 × 3) supercell to simulate the $CO_2$ activation processes, and larger p(4 × 4) supercell were utilized when considering the interactions between alkali metal cations and protons. The space above the Cu surface was filled with water molecules with a density of $1\,g\,cm^{-3}$. In order to maintain the water density, one water molecule was replaced with a $CO_2$ molecule. The volume of the solvent space was expanded with the addition of ions. Alkali metal cations $M^+$ ($Li^+$, $Na^+$, $K^+$, or $Cs^+$) were initially positioned ~5 Å away from the Cu(100) surface, while an anion $F^-$ was placed far away on the other side of the solvent to serve as an ionization pair with the cation, maintaining the applied potential close to the $CO_2$ onset potential. An additional Ne atomic layer and a 12 Å vacuum layer were added over the interface model to act as a counter electrode and monitor the electrochemical potential of the entire system[46,47].

### AIMD simulations

All AIMD simulations were performed using the Vienna Ab initio Simulation Package (VASP)[48]. The projector augmented wave (PAW) potential[49] and the generalized gradient approximation (GGA) with the revised Perdew–Burke–Ernzerhof (RPBE) functional[50] were employed to describe the electron-ion interactions and the exchange-correlation energy, respectively. The van der Waals interactions were described using the DFT-D3 correction[51]. The energy cutoff for the plane wave basis expansion was set to 500 eV. For the Brillouin zone sampling, only the gamma point was used for AIMD simulations, while a denser 3 × 3 × 1 grid was used for electronic structure calculations[52]. The Nose–Hoover thermostat was used for canonical sampling at 300 K[53,54]. The kinetic barriers and the free energy profile of the $CO_2$ activation process were studied using a combination of slow-growth[55] and blue-moon sampling[56] methods with the SHAKE algorithm[57], with the angle of the $CO_2$ molecule chosen as the collective variable (CV). The choice of CV is discussed in Supplementary Note 2.

Considering the applied potential, the Ne counter electrode[46,47] was used to monitor the potential change of the reaction system on the fly. A dipole correction[58] along the direction perpendicular to the interface was applied in the vacuum layer. The vacuum level of the metal side could be considered as constant due to the electrostatic screening of the metal, and thus the absolute potential of the system, i.e., the applied potential, could be calculated accordingly: $U$ vs SHE $= \Phi_{Interface} - 4.44\,V$, $\Phi_{Interface} = \Phi_{metal} + \frac{4\pi\mu}{A} * k$, where $\mu$ is the dipole moment of the system, $A$ is the surface area, and $k$ is Coulomb constant. Then, constant potential corrections were applied to account for the slight change in the electrochemical potential for different reaction states along the reaction paths (Supplementary Note 6)[59].

## Data availability

The data supporting the findings of this study have been included in the main text and Supplementary Information. The representative AIMD trajectories generated in this study have been deposited in the Zenodo data repository at https://doi.org/10.5281/zenodo.10427196. All other relevant data supporting the findings of this study are available from the corresponding authors upon request.

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

## Acknowledgements

This work was supported by the National Key R&D Program of China (2022YFA1203400 and 2021YFA1400100), the National Science Foundation of China (12274254 and 11874036), the Local Innovative and Research Teams Project of Guangdong Pearl River Talents Program (2017BT01N111) and Basic Research Project of Shenzhen, China (JCYJ20200109142816479 and WDZC20200819115243002). Computational resources were provided by the High-Performance Computing Platform of Nanjing University of Aeronautics and Astronautics.

## Author contributions

J.L. and F.K. conceived and designed the project. Z.Z. performed the calculation. Z.Z., J.L., H.L., Y.S., L.G., W.D. and H.H. analyzed the results. Z.Z., J.L. and F.K. wrote the paper, and all authors commented on it.

## Competing interests

The authors declare no competing interests.
