## [Peer Review File · Nature Communications]

Reviewers' comments:

Reviewer #1 (Remarks to the Author):

Please find specific comments attached.

Reviewer #2 (Remarks to the Author):

In this manuscript, the authors report the Gibbs energy barriers of the CO₂ adsorption process on Cu(100) upon the inclusion of alkali metal cations. The authors observed that the energy decreased with increasing cation size due to the coordination of the cations with the adsorbed CO₂ and the interactions with the water molecules. These results might help explain the need of alkali metal cations for the initiation of the CO₂RR observed experimentally. These are interesting results in a highly relevant topic, which could potentially be useful for understanding the underlying and complex mechanism of the CO₂RR.

Thus, I consider this work is suitable for publication in Nature Communications. Although I have some specific comments on the manuscript:

- It is not entirely clear why the angle of CO₂ was chosen as collective variable.
- What would be the effect of including a higher concentration of the alkali metal cations.
- The methodology should explain how the applied potential was accounted for.
- A short summary of previous works on CO₂RR simulations including cations should be included in the introduction.
- Would these results be expected to be independent of the metal surface/facet?
- Implicit solvation calculations are shown to yield different configurations for the adsorbed CO₂. Should these results be considered as a warning against the use of implicit solvation to account for the mechanism of electrochemical reactions? If so, this should be stated explicitly.
- The energy of the final state for the adsorbed CO₂ upon the inclusion of the cations should also be analyzed, since the trend appears different from that for the TS.
- This study explores the adsorption of CO₂ on Cu(100). The figures should explicitly state that the result corresponds to this facet and also in some points of the manuscript it should be clarified that this is the only step of the CO₂RR explored in this manuscript (the results are interesting in themselves and should not be oversell).

- It seems that when the analysis of the HER is performed only a qualitative analysis could be performed on the results. What would be a quantitative criterion to decide at what conditions there is a activation or deactivation of the CO₂ adsorption?
- Also, does the analysis for the HER reaction is considered to apply for all the other alkali metal cations different from K⁺?
- Are the simulations equilibrated? from figure 5 seems like there was a large shift in the H₂O distance just before the production steps. What was the criterion used to determine the equilibration and production steps?
- It would be useful if the authors could also include the lifetime of the hydrogen bonds with the cations.

This manuscript entitled “**Molecular Understanding of the Critical Role of Alkali Metal Cations in CO₂ Electroreduction on Copper Surface**” by Jia Li and coworkers presents a computational investigation into CO₂ adsorption onto copper surfaces. The authors present a detailed study of the process utilizing enhanced sampling methods which are time consuming and costly calculations. Below, I detail several concerns listed below that should be addressed before publication, whether in Nature Communications or elsewhere.

Specific comments:

- The calculations presented here are certainly interesting and detailed, but the novelty of the presented results is lacking. Cation effects, even specifically for the CO₂RR, have been studied using *ab initio* methods for some time now. See for example:
 - <https://doi.org/10.1039/C9EE01341E> -- which revealed the same trend that the authors see here: Li has a larger effective cation radius due to its stronger binding of its solvation shell, resulting in higher interfacial electric fields. In the case of the work submitted by the authors, this higher interfacial electric field results in a (very) slightly reduced CO₂ adsorption barrier.
 - <https://pubs.acs.org/doi/pdf/10.1021/jacs.7b06765> -- which illustrates that CO₂, when adsorbed to a metal surface, has a strong dipole moment that interacts with the changing electric field
 - <https://doi.org/10.1038/s41929-021-00655-5> -- where Nuria Lopez clearly illustrated significant changes in CO₂ binding energy with changing alkali metal cation
- Given the above, I really don't see the novelty in the presented work. Yes, the authors used enhanced sampling methods (which, granted, are extremely expensive and challenging calculations) to probe the CO₂ adsorption barrier's dependence on the alkali metal cation, but the authors find nearly no effect in this case. A change in adsorption barrier of 0.12 eV is within any reasonable DFT error estimate – a larger change might be expected just from changing the exchange correlation functional. I would guess that if the authors included 95% confidence intervals on the gathered statistics, all of the calculated barriers would be within the same confidence interval (i.e., no real calculated effect).
- Furthermore, even if a confidence interval analysis reveals a statistically significant effect, the fact of the matter remains that CO₂ adsorption is clearly not kinetically relevant on copper. Even a barrier of 0.22 eV will give rise to a turnover of 10⁹ s⁻¹.

Field Code Changed

Point-by-point responses to the reviewers' comments:

Reviewer #1:

This manuscript entitled “Molecular Understanding of the Critical Role of Alkali Metal Cations in CO₂ Electroreduction on Copper Surface” by Jia Li and coworkers presents a computational investigation into CO₂ adsorption onto copper surfaces. The authors present a detailed study of the process utilizing enhanced sampling methods which are time consuming and costly calculations. Below, I detail several concerns listed below that should be addressed before publication, whether in Nature Communications or elsewhere.

1. The calculations presented here are certainly interesting and detailed, but the novelty of the presented results is lacking. Cation effects, even specifically for the CO₂RR, have been studied using ab initio methods for some time now. See for example:

Response: We thank the reviewer for this comment, but we need to mention that certain problems and inconsistencies with experimental results remain in previous studies, including the cited references by the reviewer. For instance, the dipole-field interaction was commonly used in previous work to explain the cation size effect within the mean-field model. However, more recent experimental results have shown that the interfacial electric field is not a major contributor. In contrast, our proposed mechanism, which focuses on the Onsager reaction field strengths induced by the solvation structure, provides a comprehensive explanation for the size effect observed with alkali cations in the context of CO₂RR. Therefore, the cited references do not support the claim of novelty of our work as indicated by the reviewer, which are explained in detail as follows.

- <https://doi.org/10.1039/C9EE01341E> -- which revealed the same trend that the authors see here: Li has a larger effective cation radius due to its stronger binding of its solvation shell, resulting in higher interfacial electric fields. In the case of the work submitted by the authors, this higher interfacial electric field results in a(very) slightly reduced CO₂ adsorption barrier.

Response: We thank the reviewer for this comment, and we would like to address the points raised regarding *interfacial electric fields* and the differences between our

manuscript and the reference by Chan *et al.*

Regarding the statement on interfacial electric fields, there is a need for clarification. In the original paper by Chan *et al.*, it was indeed stated that "*cations such as Cs⁺ have the smallest hydrated cation radius and therefore show the smallest repulsion close to the electrode. The resulting higher concentrations of cations lead to a larger surface charge density and stronger interfacial electric field*". This is in contrast to the assertion in the reviewer's comment that Li⁺ was proposed to result in higher interfacial electric fields. On the other hand, it is important to note that our focus in the manuscript was not on the *dipole electric field* model. As clearly stated in paragraph 2 of the Introduction section of our manuscript (*However, this model attributed the discrepancy between different alkali cations to the tendency of larger cations to accumulate more at the interface, while ignoring the intrinsic difference of individual alkali cation*), we did not emphasize the use of interfacial electric fields to explain our results. Instead, our primary focus was on elucidating the intrinsic difference between individual alkali cations, which we believe is a key factor in understanding the observed trends. Furthermore, recent experimental results have shown that the interfacial electric field is not a major contributor to the cation size effect, and that the solvation-induced Onsager reaction field strength may be more important (JACS Au 2, 472-482 (2022)). In our manuscript, this statement was validated and, as a step forward, a molecular understanding based on the inner-sphere coordinating structures (bridge and side configurations in Figure 2 of our manuscript) between cations and intermediates *CO₂ was provided. Therefore, the main results in the paper by Chan *et al.* and in our paper are completely different, and this reference cannot cover the novelty of our work.

Furthermore, we would like to emphasize that our work differs from the reference by Chan *et al.* in two ways:

- **Explicit Solvent:** Our study incorporates explicit solvent molecules to account for the non-uniform Coulomb interactions between solvated ions and adsorbed intermediates in the electric double layer. This approach provides a more accurate representation of the system compared to the mean-field effects considered in the reference.
- **Local Interactions:** The reference by Chan *et al.* relies on a mean-field model

and does not consider local interactions, such as those between cations and their solvation shells. In our work, we observed that the key intermediate *CO_2 is involved in the solvation structure of alkali cations, which we found to be a crucial factor responsible for the cation size effect.

Therefore, we believe that the reference cited by the reviewer does not adequately support the claim that it "*revealed the same trend that the authors see here*", since our main conclusions are fundamentally different.

- <https://pubs.acs.org/doi/pdf/10.1021/jacs.7b06765> -- which illustrates that CO_2 , when adsorbed to a metal surface, has a strong dipole moment that interacts with the changing electric field.

Response: First, it is important to clarify that the dipole electric field model is not the mechanism that we propose in our work, as discussed above. Furthermore, the interactions between polarized adsorbates and interfacial electric fields, as described in this reference, are based on a uniform field model. In the original paper by Bell *et al.*, it was stated that "*The change in adsorption free energy for each species was determined by applying a uniform field oriented perpendicular to the surface in vacuum*".

Figure R1. Electric field distribution near the center of the adsorbate plotted as a function of the z-coordinate of the simulation cell for the 2^*CO initial state (From *J. Am. Chem. Soc.* **139**, 11277–11287 (2017)).

On the other hand, while it is true that the dipole of CO_2 interacts with the interfacial electric field, the estimated electric field strengths in the vicinity of the CO adsorbate with different cations do not show a consistent trend as a function of cation size (see Figure R1). This reference ultimately attributes the effect of cation size to "*an increase*

in concentration of cations at the OHP with increasing cation size," as also discussed in the above, more recent paper, by Chan *et al.* However, in our manuscript, we have ruled out the primary contribution of the electric field and identified solvation-induced local interactions as the key factor. Therefore, the results in this reference are not consistent with the statement of the reviewer, which are also significantly different from the conclusion of our manuscript.

- <https://doi.org/10.1038/s41929-021-00655-5> -- where Nuria Lopez clearly illustrated significant changes in CO₂ binding energy with changing alkali metal cation

Response: First, it is important to emphasize that in the original work by Koper *et al.* significant changes in CO₂ adsorption energy were observed when comparing systems with and without alkali cations. In particular, there was no trend, or even an opposite trend when changing between four different types of alkali cations (see Figure R2).

Figure R2. Average CO₂ adsorption Gibbs free energy at $U=0$ V versus standard hydrogen electrode (SHE) in the absence (grey) or presence (light to dark brown) of an alkali cation (From *Nat. Catal.* **4**, 654-662 (2021)).

In fact, the absence of a clear trend is quite reasonable, since the influence of variations in binding energies on the kinetic barrier and consequently on the activity remains uncertain. It's worth noting that our computational results have also shown that the adsorption energies of CO₂ do not show a pronounced trend (see Supplementary Table 1 in our manuscript). Therefore, CO₂ adsorption energy alone cannot be considered as a reliable activity descriptor. In light of these considerations, we extended our research by employing enhanced sampling methods to assess the activation barrier in the presence of different cations. This approach allowed us to successfully identify the increasing activity of the CO₂ reduction reaction ranging from pure water to smaller

Li⁺/Na⁺ cations and extending to larger K⁺/Cs⁺ cations.

In summary, it is important to emphasize that the comments raised here are not consistent with the primary conclusions of our study, and the cited references do not support the claim of novelty of our work as indicated by the reviewer. Therefore, we respectfully ask the reviewer to reconsider the comment of "*the novelty of the presented the results is lacking*".

2. Given the above, I really don't see the novelty in the presented work. Yes, the authors used enhanced sampling methods (which, granted, are extremely expensive and challenging calculations) to probe the CO₂ adsorption barrier's dependence on the alkali metal cation, but the authors find nearly no effect in this case. A change in adsorption barrier of 0.12 eV is within any reasonable DFT error estimate – a larger change might be expected just from changing the exchange correlation functional. I would guess that if the authors included 95% confidence intervals on the gathered statistics, all of the calculated barriers would be within the same confidence interval i.e., no real calculated effect).

Response: We thank the reviewer for this comment. And we would like to address the concerns and provide further clarification.

Regarding the novelty of our work, we understand that the reviewer is trying to distinguish our manuscript from the referenced works. As we mentioned in our previous responses to Comment 1, we believe that the novelty lies in the specific focus of our study, which examines the influence of alkali metal cations on the CO₂ adsorption barrier using enhanced sampling methods. While we acknowledge that the change in the adsorption barrier (0.12 eV) is within the range of typical DFT error estimates, we would like to emphasize that our study goes beyond the thermodynamic binding energy and focuses on the kinetic barrier. This kinetic calculation, within the framework of *ab initio* molecular dynamics, requires the use of enhanced sampling methods to accurately capture the dynamic behavior. The choice of enhanced sampling methods is a crucial aspect of our research, and it arises from the need to account for local interactions within the solvation shell of alkali cations.

Regarding error estimation, we would like to clarify that we have indeed taken steps to

evaluate the errors in our simulations. We chose to use the blue-moon method instead of the slow-growth method to perform the thermodynamic integration. This is simply because the standard error can be evaluated well in the blue-moon method. As explained in Supplementary Note 3 of our manuscript, we used block averages to analyze the errors at each point of the barrier calculations, and the standard errors were integrated to generate error bars for the free energy barrier calculations. These error bars are presented in Figure 3 of our manuscript. Importantly, these error bars do not obscure the observed trend in our data, suggesting that the effect we report is not merely within the bounds of statistical variation.

We acknowledge that the absolute values may vary with different exchange correlation functional, as mentioned by the reviewer. However, our primary focus is on the general trend and underlying molecular explanations derived from a comprehensive and carefully evaluated simulation framework. Reproducing the exact experimental results can be challenging due to the complexity of real systems. There are so many possible factors that could affect the current density in reality, such as the CO₂ solubility. However, the magnitude of the changes in the rate constant resulting from the change in the adsorption barrier of 0.12 eV compare reasonably well with the experimentally determined variations in current density. And the trend between the two cation groups, smaller Li⁺/Na⁺ and larger K⁺/Cs⁺, could already well explain the differences in Onsager electric field strengths in our manuscript. And we believe that this trend and the underlying molecular explanation could be helpful in understanding the interfacial dynamics during electrocatalysis.

3. Furthermore, even if a confidence interval analysis reveals a statistically significant effect, the fact of the matter remains that CO₂ adsorption is clearly not kinetically relevant on copper. Even a barrier of 0.22 eV will give rise to a turnover of 10⁹ s⁻¹.

Response: We thank the reviewer for this comment. We agree that a barrier of 0.22 eV could still result in high activity, which cannot explain the necessity of alkali cations. Therefore, to address this issue, we further investigated the possible influence of alkali cations on proton accessibility and competing HER activity on the copper surface (Section 2.3 Necessary Effect of M⁺ on CO₂RR in the manuscript). We observed that alkali metal cations repel protons away from the interface and confine them in the

solvation shell, thereby inhibiting the reactivity of the competing HER and allowing CO₂ reduction to initiate. Therefore, our results provide new insights into the design of electrochemical environments and highlight the importance of explicitly including the solvation and competing reactions in a comprehensive understanding of cation effects on CO₂RR.

Reviewer #2:

In this manuscript, the authors report the Gibbs energy barriers of the CO₂ adsorption process on Cu(100) upon the inclusion of alkali metal cations. The authors observed that the energy decreased with increasing cation size due to the coordination of the cations with the adsorbed CO₂ and the interactions with the water molecules. These results might help explain the need of alkali metal cations for the initiation of the CO₂RR observed experimentally. These are interesting results in a highly relevant topic, which could potentially be useful for understanding the underlying and complex mechanism of the CO₂RR.

Thus, I consider this work is suitable for publication in Nature Communications. Although I have some specific comments on the manuscript:

1. It is not entirely clear why the angle of CO₂ was chosen as collective variable.

Response: We thank the reviewer for this comment. When considering CO₂ adsorption, two parameters can be chosen as collective variables. One is the possibly more commonly used bond length of the Cu-C bond (C is from CO₂) and the other is the CO₂ angle. Through extensive MD simulations for initial states (physisorbed CO₂ with flat configuration) and final states (chemisorbed CO₂ with bent configuration) with different cations, we have gained insight into the main factors influencing the difference between the initial states and the final states. Our results indicate that on the Cu(100) surface, the CO₂ angle is relatively more robust than the Cu-C bond length, making it a more suitable choice as the collective variable to describe the reaction coordinates.

For the initial states with physisorbed CO₂, the CO₂ molecule exhibited translational and rotational motions, positioning itself approximately 3.4 Å away from the interface, influenced by the dynamics of the surrounding water molecules. Consequently, the determination of the specific Cu atom where the CO₂ molecule can adsorb became challenging due to the weak interaction. However, the CO₂ angle consistently shows fluctuations around an average of about $175^\circ \pm 3^\circ$ with a Gaussian distribution (see Supplementary Figure 4). Notably, this is different from the single atom catalyst, where the adsorption site is predetermined, allowing for the selection of a simple constraint, such as the C-M bond, as a collective variable.

Furthermore, our analysis of the adsorption configuration of chemisorbed CO₂ in the final states further supports our choice of the CO₂ angle as the collective variable. In this configuration, bent CO₂ adsorbs on the Cu(100) surface via both Cu-C and Cu-O bonds, with the C atom positioned at the bridge site, as opposed to the top site traditionally considered in implicit solvation. This implies that a single Cu-C bond alone cannot fully capture the achievement of the final state. However, the CO₂ angle consistently follows a Gaussian distribution centered around approximately 119° ± 3°, providing a robust description of the reaction coordinate.

Change to manuscript: We have added the description of the collective variable in Supplementary Note 2 of the Supplementary Information.

2. What would be the effect of including a higher concentration of the alkali metal cations.

Response: We thank the reviewer for this comment. The effect of increasing the concentration of alkali metal cations is a multifaceted issue influenced by various factors such as cation concentration, surface charge density, applied potential, intermediate coverage, and more. While the full complexity of this issue is beyond the scope of the discussion in our manuscript, we can offer some insights into this issue.

Figure R3. (a) Averaged Li⁺-Li⁺ distance in the system with two Li⁺ cations and one *CO₂ at the interface between the Cu(100) surface and the electrolyte. An exemplified snapshot is shown in (b).

If we limit our analysis to a simulation cell containing a higher concentration of M⁺ ions (e.g., two M⁺ cations) with a single *CO₂ molecule, we can expect the effects of

individual alkali cations to be amplified at higher concentrations. This is because solvated cations tend to coordinate with *CO₂ to form M⁺-*CO₂ complexes. In addition, solvated cations will repel each other to maintain the integrity of their solvation shells. For example, in a system with two Li⁺ cations, the average Li⁺-Li⁺ distance is about 6.5 Å (as shown in Figure R3), which exceeds the diameter of the first solvation shell of Li⁺.

In this context, it is reasonable to assume that the interaction of multiple cations with a single *CO₂ molecule is not significantly different from that of a single M⁺-*CO₂ complex. Therefore, the primary effect of a higher cation concentration is the increased formation of M⁺-*CO₂ pairs, which in turn promotes the CO₂ reduction reaction (CO₂RR). It is important to note that higher cation concentration typically corresponds to higher surface charge density and lower applied potential. These factors also influence the electron transfer-coupled CO₂ activation process, adding further complexity to this issue.

However, if we consider variations in the equilibrated interfacial concentrations of different cations under the same applied potential, the issue becomes much more complicated and difficult to address by simulation. While a common hypothesis is that the same applied potential on a specific substrate will result in the same surface charge density and require a roughly equivalent amount of cations/anions to neutralize the charges. However, in a real system, the cation concentration at the interface may differ from that in the bulk electrolyte, and the electrochemical double layer (EDL) response of different alkali cations may also vary due to differences in capacitance. In addition, the exact relationship or ratio between bulk and interfacial concentrations remains unknown. Consequently, determining the appropriate number of ions to use for different alkali cations at specific potentials is a challenging task, making direct comparisons between multiple cations infeasible. Therefore, our work has focused primarily on investigating the individual effects of alkali cations.

3. The methodology should explain how the applied potential was accounted for.

Response: We thank the reviewer for this constructive suggestion. Considering the applied potential, the Ne counter electrode was used to monitor the potential change of the reaction system on-the-fly. A dipole correction along the direction perpendicular to

the interface was applied in the vacuum layer. The vacuum level of the metal side could be considered as constant due to the electrostatic screening of the metal, and thus the absolute potential of the system, i.e., the applied potential, could be calculated accordingly: $U \text{ vs. SHE} = \Phi_{\text{Interface}} - 4.44 \text{ V}$, $\Phi_{\text{Interface}} = \Phi_{\text{metal}} + \frac{4\pi\mu}{A} * k$, where μ is the dipole moment of the system, A is the surface area, and k is the Coulomb constant. Constant potential correction was then applied to account for the slight change in the electrochemical potential for different reaction states along the reaction path.

Change to manuscript: We have added a brief description to discuss how the applied potential is considered in the *Methods* section of the manuscript:

*"Considering the applied potential, the Ne counter electrode was used to monitor the potential change of the reaction system on-the-fly. A dipole correction along the direction perpendicular to the interface was applied in the vacuum layer. The vacuum level of the metal side could be considered as constant due to the electrostatic screening of the metal, and thus the absolute potential of the system, i.e., the applied potential, could be calculated accordingly: $U \text{ vs. SHE} = \Phi_{\text{Interface}} - 4.44 \text{ V}$, $\Phi_{\text{Interface}} = \Phi_{\text{metal}} + \frac{4\pi\mu}{A} * k$, where μ is the dipole moment of the system, A is the surface area, and k is the Coulomb constant. Then, constant potential corrections were applied to account for the slight change in the electrochemical potential for different reaction states along the reaction paths (Supplementary Note 6)."*

And the detailed explanations have been added in Supplementary Note 1 and Note 6.

4. A short summary of previous works on CO₂RR simulations including cations should be included in the introduction.

Response: We thank the reviewer for this valuable suggestion. We have added a brief summary of previous work, mainly on explicit solvation and AIMD simulations, to the *Introduction* section of the manuscript,

"... These findings suggest that different alkali cations have inherent differences rather than a simple concentration disparity, and that cations at the interface could greatly influence the efficiency of the CO₂ to CO conversion process, potentially playing a more active role rather than merely acting as spectators. To provide a more detailed explanation of the influence of solvated cations on reaction mechanisms, AIMD simulations were performed and revealed the existence of local interactions, such as

*coordination between solvated cations and surface intermediates. Notable examples include the interaction between K^+ cation and adsorbed $*CO_2$ molecule on Au surface, and the interaction between $Li^+/K^+/Cs^+$ cations and $*CO+*CO$ dimer on Cu surface. However, a comprehensive understanding of the size effect of alkali cations on CO_2 activation and their critical role in CO_2RR initialization on Cu surface is still lacking."*

5. Would these results be expected to be independent of the metal surface/facet?

Response: We thank the reviewer for raising this issue. We believe that some of the conclusions of our work could be extended to other metal surfaces/facets. For example, the promoting role of the cation originating from the M^+*CO_2 complex could potentially be expected in similar systems. However, it is important to note that the specific adsorption configurations may differ from those observed on Cu(100) due to variations in substrate properties, which is an area of great interest for our future research.

Basically, the effect of different metal surfaces or facets on an electrochemical system can be divided into two main aspects. First, it can alter the active sites, thereby affecting the binding strengths between the substrate and intermediates due to changes in substrate properties or the coordination number of the active sites. Second, it can alter the potential of the zero charge, thus influencing the surface charge density under specific applied potentials. Since the activation of CO_2 involves electron transfer processes, changes in the surface charge density can significantly affect the driving forces provided by the applied potentials, potentially leading to observable phenomena. For example, in our simulations we observed stable adsorption of bent $*CO_2$ on Cu(100) surfaces without any constraints or external factors such as cations or potentials. In contrast, previous reports on the Au(111) surface indicated that bent $*CO_2$ could not be stably adsorbed even in the presence of K^+ ions (Nat Catal 5, 977-978 (2022)). However, it is further demonstrated that bent $*CO_2$ can become a stable configuration at higher surface charge densities on the Au(110) surface (J. Am. Chem. Soc. 145, 3, 1897-1905 (2023)).

Despite the different stable adsorption configurations observed on different metal surfaces, the promoting role of alkali cations remains a consistent factor. Therefore, we expect that some of the findings from our work can be applied to other metal surfaces

and facets, and that additional effects will also be explored in our future work.

Change to manuscript: We have added a brief discussion part in the last paragraph of section 2.2 of the manuscript:

*"Furthermore, similar behavior could also be expected on other coinage metal surfaces or facets due to their similar response to alkali cations. Basically, the effect of different metal surfaces or facets on an electrochemical system can be divided into two main aspects. First, it can alter the active sites, thereby affecting the binding strengths between the substrate and intermediates due to changes in substrate properties or the coordination number of the active sites. Second, it can alter the potential of the zero charge, thus influencing the surface charge density under specific applied potentials. Since the activation of CO₂ involves electron transfer processes, changes in the surface charge density can significantly affect the driving forces provided by the applied potentials, potentially leading to observable phenomena. For example, in our simulations we observed stable adsorption of bent *CO₂ on Cu(100) surfaces without any constraints or external factors such as cations or potentials. In contrast, previous reports on the Au(111) surface indicated that bent *CO₂ could not be stably adsorbed even in the presence of K⁺ ions. However, it is further demonstrated that bent *CO₂ can become a stable configuration at higher surface charge densities on the Au(110) surface. Despite the different stable adsorption configurations observed on different metal surfaces, the promoting role of alkali cations remains a consistent factor. Therefore, we expect that some of the findings from our work can be applied to other metal surfaces and facets, and that additional effects will also be explored in our future work."*

6. Implicit solvation calculations are shown to yield different configurations for the adsorbed CO₂. Should these results be considered as a warning against the use of implicit solvation to account for the mechanism of electrochemical reactions? If so, this should be stated explicitly.

Response: We thank the reviewer for this valuable suggestion. In fact, we have further evaluated the possible adsorption configurations in implicit solvent, i.e., considering the physisorbed linear CO₂, the chemisorbed *CO₂ at the atop site with a single C-Cu bond formation and both oxygen atoms symmetrically facing upward, and the bidentate configuration. Using the grand canonical potential method, we surprisingly found that

the physisorbed linear CO₂ is indeed dominant at a potential as low as -1.8 V_{SHE} in implicit solvent, which has reached the CO₂RR active region (Figure R4). In particular, the bidentate configuration could be energetically less stable than the physisorbed CO₂ by more than 0.3 eV at the potential of -0.6 V_{SHE}, which is the concerned potential range in our manuscript. This difference obviously indicates the crucial role of hydrogen bond in stabilizing *CO₂ at Cu(100), and thus raises the need to re-evaluate previously proposed mechanisms, especially those involving interactions between intermediates and the surrounding electrochemical environment under implicit solvation conditions. While thermodynamic simulations can still provide approximate estimates, a careful investigation of the reaction mechanisms at the molecular level is indispensable.

Figure R4 | Stable adsorption configurations of CO₂@Cu(100) in fully implicit solvent. (a) Under constant charge condition, the comparison of total energies between three representative adsorption configurations shown as illustrations, where blue, red and brown spheres represent Cu, O and C atoms, respectively. (b) Under constant potential condition, the comparison of grand canonical potentials between these three systems.

Change to manuscript: To address this concern, we have strengthened our discussion of this issue in the manuscript, specifically in paragraph 3 of Section 2.1,

*"Furthermore, it is worth noting that the bidentate configuration of *CO₂ on the Cu(100) surface differs from that observed in a fully implicit solvent environment. In the latter case, CO₂ tends to be physisorbed on the Cu(100) surface over a wide potential range, while the bidentate configuration is less stable by ~0.3 eV at -0.6 V_{SHE}, which is the concerned potential range in this work (Supplementary Figure 3). This difference indicates the significant contribution of surrounding hydrogen bonds to the stabilization of *CO₂ from explicit solvation, and thus raises a cautionary note regarding the use of fully implicit solvents for investigating molecular explanations of*

reaction mechanisms, especially when the interactions between intermediates and the surrounding electrochemical environment, such as solvated water molecules, play a substantial role."

Figure R4 has been added in the Supplementary Information as Supplementary Figure 3.

7. The energy of the final state for the adsorbed CO₂ upon the inclusion of the cations should also be analyzed, since the trend appears different from that for the TS.

Response: We thank the reviewer for this comment. The calculated adsorption free energy (ΔG_{ads}) shows the lowest value with Li⁺, while ΔG_{ads} with the other three cations do not differ much from each other. This is consistent with previous results on the Au(111) surface, where no obvious trend of adsorption energy could be observed when changing the alkali cations (Figure R2, Nat. Catal. 4, 654-662 (2021)). This could be reasonable since the reaction free energy is not always strongly correlated with the kinetic barrier, while the latter is the true rate-determining factor. Therefore, we attributed the cation effects to originated from their influences on the activation process, i.e., from the initial state to the transition state, and concentrated mainly on the analysis of the transition state and the activation barrier of CO₂ with different cations.

Change to manuscript: We have added the above discussions in paragraph 2 of Section 2.2 of the manuscript,

*"... First, we considered the CO₂ adsorption free energy with different cations as an approximation of the energetic descriptor. The system with Li⁺ shows the strongest adsorption of *CO₂, while the adsorption free energies with the other three cations do not differ much from each other (Fig. 3 and Supplementary Table 3). Consequently, no clear trend with cation size could be observed, which is also consistent with previous results on Au(111). As repeatedly demonstrated, the reaction energy alone cannot fully describe the activity of a system, while the activation energy from initial state (IS) to transition state (TS) might be more important to serve as the real energetic criteria. Therefore, in order to obtain more accurate and detailed information about the cation effects on the CO₂ adsorption process, we next concentrated on the free energy barrier and transition states of the CO₂ adsorption process near the onset potential of CO₂RR at Cu(100)..."*

8. This study explores the adsorption of CO₂ on Cu(100). The figures should explicitly state that the result corresponds to this facet and also in some points of the manuscript it should be clarified that this is the only step of the CO₂RR explored in this manuscript (the results are interesting in themselves and should not be oversell).

Response: We thank the reviewer for this comment. We have re-checked the manuscript and revised the relevant expressions in the manuscript, and added a statement at the beginning of Section 2.2 explaining that we are focusing only on the CO₂ adsorption process during CO₂RR,

"...CO₂RR is inhibited in the early stages of the CO₂ to CO conversion process. To investigate the underlying mechanism, here we first focused on the effect of alkali metal cations on the rate-limiting step of CO₂RR to CO, the CO₂ activation process, which is expected to have electrostatic interactions with alkali cations due to its transformation from a linear non-polar configuration to a bent polar one..."

9. It seems that when the analysis of the HER is performed only a qualitative analysis could be performed on the results. What would be a quantitative criterion to decide at what conditions there is an activation or deactivation of the CO₂ adsorption?

Response: We thank the reviewer for raising this issue. However, it is challenging to adequately address in our current AIMD simulations. Typically, we rely on differences in activation barriers to provide a quantitative criterion for determining the main reaction pathway. In this particular case, however, the inhibition of proton approach to the surface by cations emerges as the critical factor. This means that the concentration of reactants, influenced by both potential, ionic activities, and mass transport, may play the dominant role. At the same time, establishing a precise quantitative relationship between interfacial reactant concentration and cation properties in AIMD simulations proves to be challenging. Therefore, we regret to admit that our simulation can only provide qualitative trends, and the establishment of a quantitative criterion in this context remains elusive.

10. Also, does the analysis for the HER reaction is considered to apply for all the other alkali metal cations different from K⁺?

Response: We thank the reviewer for this comment. We have performed additional

calculations to investigate the effect of other alkali metal cations on the HER reaction. Figure R5 shows that similar inhibition effects of K^+ on HER were also obtained for Na^+ and Cs^+ , as indicated by the increased proton-interface distances with these three types of alkali cations. Moreover, the comparable proton-interface distances and the diameters of the first solvation shells of alkali cations suggested that these cations could repel the proton away from the interface and confine the proton in their first solvation shells. Note that the average proton-interface distance for Cs^+ is slightly smaller than those for Na^+ and K^+ , due to the partial desolvation behavior of $Cs(H_2O)_x$. In the case of Li^+ , however, the proton-interface distance seems to be even smaller than in the case of a single H_3O^+ , which could be due to two aspects. On the one hand, the solvation shell of Li^+ is smaller than that of the other alkali cations, which means that at a relatively low concentration, the H_3O^+ and the solvated Li^+ could still coexist near the interface despite of the electrostatic repulsion between them. On the other hand, after introducing alkali cations into the systems, the negative charge density at the Cu(100) surface is increased compared to that of single H_3O^+ , which further attracts the proton to approach the interface. These two aspects lead to the abnormal decrease of the proton-interface distance with Li^+ cation. Therefore, from the analysis of the interactions between protons and alkali cations, we can find that the inhibition effect of alkali cations on HER is also correlated with the cation size, which would further affect the selectivity of CO_2RR vs. HER combined with the size-dependent promoting effect on CO_2 activation.

Figure R5 | Interaction between solvated proton (H_3O^+) and alkali cations. (a) Distribution of H_3O^+ in the solvent with and without alkali cations. The red and purple dashed lines represent the average distance of the excess proton from the Cu surface in the production timescale of 10 ps with and without K^+ cation, respectively. The corresponding representative interface structures are shown

in (b), where H_3O^+ and K^+ are highlighted shown using ball and stick model and other atoms are hidden for clarity.

Change to manuscript: We have revised the discussions of the interactions between all four types of alkali cations with the proton in Section 2.3 of the manuscript,

*"Figure 5 shows the distributions of an excess proton with and without the alkali cations (taking the K^+ cation as an example) at the interface. Cumulative averages of the results are shown in **Supplementary Figure 9**. The excess proton is solvated to form a hydronium ion (H_3O^+), which can move along the hydrogen bond network by hopping between different water molecules. After equilibrating for more than 20 ps, the solvated proton reaches a dynamically stable configuration at a distance of about 5.2 Å away from the Cu surface. Based on the water distribution shown in **Fig. 1**, the excess proton diffuses into the second water layer (third peak in **Fig. 1a**), which satisfies the three hydrogen bonds required by the H_3O^+ ion. However, when alkali cations are introduced at the interface, the proton-interface distance increases, except for Li^+ (**Fig. 5a**). The change in proton-interface distance results from the disruption of the interfacial hydrogen bond network and the electrostatic repulsion of solvated alkali cations. For larger cations, including Na^+ , K^+ , and Cs^+ , the excess proton is repelled away from the second water layer and confined in the first solvation shells of these cations, as indicated by the comparable proton-interface distances and the diameters of the first solvation shells of alkali cations. The average proton-interface distance with Cs^+ is slightly less than twice the radius of the first solvation shell due to its partial desolvation behavior, similar to the case in **Supplementary Figure 7**. As for the case with Li^+ , the proton-interface distance seems to be even smaller than in the case with a single H_3O^+ , which is due to two aspects. On the one hand, the solvation shell of Li^+ is smaller and harder than that of larger alkali cations, as indicated in the previous sections, which means that the interfacial hydrogen bond network is relatively more complete, leaving space for H_3O^+ to form three required hydrogen bonds at the interface. In this way, the H_3O^+ and solvated Li^+ could coexist between the first and second water layers despite of the electrostatic repulsion between them. Furthermore, after the introduction of alkali cations, the negative charge density at the Cu(100) surface increases compared to that of single H_3O^+ , which further attracts the proton to approach the interface. These two aspects result in the abnormal decrease of the*

proton-interface distance between the proton and the Li⁺ cation at the interface. Furthermore, the hydrogen bond lifetime decreases from 3.62 ps to 2.44 ps with the inclusion of the Li⁺ cation, and is further suppressed with increasing cation size to only 1.05 ps after introducing Cs⁺, which provide additional support for the influence of solvated alkali cations on solvent dynamics.

Note that we are only showing the qualitative trend here, since in reality the interfacial concentration of alkali cations under a specific potential could also deviate from each other due to the different capacitance responses, but the concentration change can hardly be reflected in simulations at the nanoscale, since only few ions can be introduced due to the scale limitation. However, we believe that the repulsion and confinement effects between the interfacial proton and the alkali cations exist, especially for larger ones, which could strongly inhibit the transport of the proton from the bulk electrolyte to the interface, and thus suppress the activity of HER..."

And we have replaced Figure 5 with Figure R5 in the manuscript.

11. Are the simulations equilibrated? from figure 5 seems like there was a large shift in the H₂O distance just before the production steps. What was the criterion used to determine the equilibration and production steps?

Response: We thank the reviewer for this comment. The equilibration criterion used in our study involves monitoring the total energy of the system and the motion of the proton to ensure their smooth fluctuations around a stable value for a duration greater than 5 ps. Figure R4 shows that a single proton reaches equilibration relatively quickly, stabilizing in the range of 10 to 15 ps. In contrast, the interaction between K⁺ and H⁺ ions takes about 20 ps to reach equilibration. We have also performed additional simulations for the latter case, which do not show significant deviations from the observed trends.

For clarity and completeness, we have made updates to Figure 5a, as shown in our response to the above comment. These updates extend the production time for the K⁺+H⁺ case and include results obtained with other alkali cations.

12. It would be useful if the authors could also include the lifetime of the hydrogen bonds with the cations.

Response: We thank the reviewer for this valuable suggestion. To evaluate the lifetime of hydrogen bond in systems containing different alkali cations and/or a proton, we selected the trajectories from the last 10 ps of each simulation as a representative sample. An example result with K^+ is shown in Figure R6, and all calculated hydrogen bond lifetimes are shown in Table R1, where detailed methods described below Table R1. In general, the calculated hydrogen bond lifetimes are shorter than those found in bulk liquid water (Chem. Sci., **9**, 2065-2073 (2018), J. Phys. Chem. B **117**, 50, 16188–16195 (2013)). This discrepancy may be due to the perturbation of the hydrogen bond network by the solid-liquid interface and the constraints imposed by the size of the simulation cell. Furthermore, the hydrogen bond lifetime decreases with the inclusion of alkali cations and is further suppressed with increasing cation size, highlighting the influence of solvated alkali cations on solvent dynamics.

Figure R6. Hydrogen bond time autocorrelation functions and fitted results with K^+ and H^+ cations at the Cu(100)-electrolyte interface. Dashed lines indicate the original data, while solid lines show the fitted curves. Red and blue colors correspond to continuous and intermittent hydrogen bond time autocorrelation functions, respectively.

Table R1. Hydrogen bond lifetime with different cations

System	Continuous H-bond lifetime $\tau_c(ps)$	Intermittent H-bond lifetime $\tau_I(ps)$
H^+	0.19	3.62
$H^+&Li^+$	0.16	2.44
$H^+&Na^+$	0.16	2.17
$H^+&K^+$	0.13	1.45
$H^+&Cs^+$	0.12	1.05

Note: The hydrogen bond lifetime is estimated using the time autocorrelation function;

$$C(t) = \left\langle \frac{\sum h_{ij}(t_0) h_{ij}(t_0 + \tau)}{\sum h_{ij}(t_0)^2} \right\rangle$$

Here, h_{ij} indicates the presence of a hydrogen bond between atoms ij : if a hydrogen bond exists, $h_{ij} = 1$; otherwise, $h_{ij} = 0$. τ is the time variable to determine whether a specific hydrogen bond still exists. t_0 is the time origin, and the average (indicated by angular bracket) is taken over a series of time origins with an interval of Δt , where Δt is chosen to be 1 ps. The results of the autocorrelation functions were fitted with the biexponential function, $C(t) = A \exp(-t/\tau_1) + B \exp(-t/\tau_2)$, and the average hydrogen bond lifetime is determined as $A\tau_1 + B\tau_2$. In addition, intermittent hydrogen bond lifetimes are also calculated with a tolerant time of 0.2 ps. This allows a hydrogen bond to be considered present even if it breaks up to the specified time scale.

Change to manuscript: We have added the description of hydrogen bond lifetime in paragraph 2 of Section 2.3 of the manuscript,

"...Furthermore, the hydrogen bond lifetime decreases with the inclusion of the Li⁺ cation, and is further suppressed with increasing cation size (Supplementary Note 5), providing additional support for the influence of solvated alkali cations on solvent dynamics. ..."

And the methods and results about the hydrogen bond lifetime are added in Supplementary Note 5 of the Supplementary Information.

REVIEWER COMMENTS

Reviewer #1 (Remarks to the Author):

Please find comments in the document attached.

Reviewer #2 (Remarks to the Author):

The authors sent a revised version of the manuscript NCOMMS-23-23847. The authors carefully responded to all the queries and included additional calculations to support their arguments. The authors responded to all the questions I addressed in the initial review of this work, and I consider the manuscript to be suitable for publication in Nature Communications. I have only a few minor comments that I believe should be addressed beforehand:

- The title of the manuscript is misleading since the results presented refer to the CO₂ adsorption process on Cu(100) at -0.6 V vs SHE and not CO₂ electroreduction in general.
- The reflection presented in the reply letter regarding the elusiveness of a quantitative criterion in their simulations of the HER should be included in the discussion of the manuscript.
- This phrase can be a bit confusing "These two aspects result in the abnormal decrease of the proton-interface distance between the proton and the Li⁺ cation at the interface." The authors should try to write it in a clearer way.
- The plots of the total energy of the system as function of time should be included in the SI to show that indeed the systems are equilibrated. The reply letter refers to an incorrect plot and this information was not presented.
- Referring to the differences among the cations as size effects should be used with more caution as this could probably be associated with their only difference being their size (the word effect is my main concern is with the word effect in this case). This and the fact that only the effect of alkali metals was analyzed should be clarified.

The revised manuscript has significantly improved the clarity of the submitted work, particularly in distinguishing the work here with mean-field approaches from Chan et al and Nørskov et al. A number of interesting conclusions come from this manuscript; however, it is not clear to me that the insights gained from this work meet the high bar for publication in this journal. After carefully reading the author's rebuttal and revised manuscript, the two primary insights derived from the manuscript (according to this reviewer) are summarized as follows:

1. In contrast to prior theoretical/computational investigations into the CO₂RR on coinage metals, and in support of at least one recently reported *operando* spectroscopic study, alkali metal cations enhance CO₂ adsorption not by a direct dipole-field interaction, but by enhancing direct coordination with the final state of the first step (i.e. chemisorbed CO₂). This has been previously reported by Koper et al as I suggested in my prior review. The new advancement in this work is to extend the analysis to the activation barrier for CO₂ adsorption using enhanced sampling methods. Although technically sound and interesting, the result is unfortunately kinetically irrelevant as the authors note. The fact that CO₂ adsorption seems to be kinetically irrelevant for the process could itself be an interesting finding from this work, though the authors do not suggest this in their analysis.
2. Since there is (apparently) no kinetically relevant effect with CO₂ itself, the authors investigate the role that alkali metal cations play on the primary side-reaction during the CO₂RR, i.e. HER. Here, the authors find that the presence of cations such as K⁺ and Cs⁺ results in hydronium equilibrium distances significantly farther from the metal surface; for potassium, about a 1.6Å difference is noted, similar in magnitude to the effect reported for sodium ions in a recent investigation of pH effects in the HER (<https://doi.org/10.1038/s41929-022-00846-8>). In that work (uncited by the present manuscript), the authors similarly conclude that a disrupted H-bond network introduced by cations may be a causal mechanism for decreased HER activity. So, the new insight for the submitted work here is a demonstration of this phenomenon extending to copper, and identification of a possible trend with cation size while the prior reports typically focus on just a single cation.

Given the above, to maximize impact and novelty of the work, I would suggest that the authors reorganize their manuscript to instead focus on the effect of cations on the HER, since it seems that their work suggests there is not a significant effect on activation of CO₂ itself. Other findings in the manuscript (e.g., purely implicit solvation failing to find the correct binding configuration of CO₂), while valuable, are unlikely to be of interest outside of niche groups.

Point-by-point responses to the reviewers' comments:

Reviewer #1:

The revised manuscript has significantly improved the clarity of the submitted work, particularly in distinguishing the work here with mean-field approaches from Chan et al and Nørskov et al. A number of interesting conclusions come from this manuscript; however, it is not clear to me that the insights gained from this work meet the high bar for publication in this journal. After carefully reading the author's rebuttal and revised manuscript, the two primary insights derived from the manuscript (according to this reviewer) are summarized as follows:

1. In contrast to prior theoretical/computational investigations into the CO₂RR on coinage metals, and in support of at least one recently reported *operando* spectroscopic study, alkali metal cations enhance CO₂ adsorption not by a direct dipole-field interaction, but by enhancing direct coordination with the final state of the first step (i.e. chemisorbed CO₂). This has been previously reported by Koper et al as I suggested in my prior review. The new advancement in this work is to extend the analysis to the activation barrier for CO₂ adsorption using enhanced sampling methods. Although technically sound and interesting, the result is unfortunately kinetically irrelevant as the authors note. The fact that CO₂ adsorption seems to be kinetically irrelevant for the process could itself be an interesting finding from this work, though the authors do not suggest this in their analysis.

Response: We thank the reviewer for this comment. As highlighted in our previous response, while the magnitude of the energy barrier change in the rate constant is in reasonable agreement with the experimentally determined variations in current density, reproducing the exact experimental results remains challenging due to the complexity of real systems. Therefore, our primary focus is on the general trend and molecular differences among the alkali metal cations, which have been already well explained in our manuscript based on the differences in solvation structures and Onsager field strengths. We acknowledge that the possible coordination between chemisorbed CO₂ and K⁺ cation has been proposed by Koper et al, but emphasize that the general differences among alkali metal cations have not been adequately addressed in their work.

Furthermore, we advocate a systematic exploration of the effects of alkali metal cations on CO₂ activation and competing HER on the copper surface. This exploration is not only highly demanded but also has the potential to inspire future studies. For example, why CO₂ cannot be stably chemisorbed on Au(110) and Au(100) surfaces without alkali metal cations (Nat. Commun. **14**, 7607 (2023), J. Am. Chem. Soc. **145**, 3, 1897-1905 (2023), Nat Catal **5**, 977-978 (2022)), while the cases are completely different on Cu(100) surface, where the necessary effect of alkali metal cations should consider CO₂ activation and HER inhibition. To address such a complex issue, we emphasize the necessity of a comprehensive molecular understanding of all relative reaction pathways through kinetic simulations, highlighting the importance of the underlying molecular explanation in understanding the interfacial dynamics during electrocatalysis, in addition to the energetic values.

2. Since there is (apparently) no kinetically relevant effect with CO₂ itself, the authors investigate the role that alkali metal cations play on the primary side reaction during the CO₂RR, i.e. HER. Here, the authors find that the presence of cations such as K⁺ and Cs⁺ results in hydronium equilibrium distances significantly farther from the metal surface; for potassium, about a 1.6Å difference is noted, similar in magnitude to the effect reported for sodium ions in a recent investigation of pH effects in the HER (<https://doi.org/10.1038/s41929-022-00846-8>). In that work (uncited by the present manuscript), the authors similarly conclude that a disrupted H-bond network introduced by cations may be a causal mechanism for decreased HER activity. So, the new insight for the submitted work here is a demonstration of this phenomenon extending to copper, and identification of a possible trend with cation size while the prior reports typically focus on just a single cation.

Response: We thank the reviewer for this comment and for recognizing the crucial conclusion regarding the distinct effect of alkali metal cations on the HER in our manuscript. However, we would like to clarify that the reference mentioned by the reviewer was actually cited in our original manuscript as Ref. 37. In addition, we emphasize that a comprehensive understanding of the effects of alkali metal cations on the CO₂RR at the copper surface necessitates considering their effects on both CO₂ activation and proton availability, as we stated in our previous response.

Given the above, to maximize impact and novelty of the work, I would suggest that the authors reorganize their manuscript to instead focus on the effect of cations on the HER, since it seems that their work suggests there is not a significant effect on activation of CO₂ itself. Other findings in the manuscript (e.g., purely implicit solvation failing to find the correct binding configuration of CO₂), while valuable, are unlikely to be of interest outside of niche groups.

Response: We thank the reviewer for this suggestion. After careful consideration of both the reviewer's and the editor's suggestions, we believe that the current organization of our manuscript effectively presents a comprehensive exploration of the effects of alkali metal cations on CO₂ reduction at the Cu(100) surface. This exploration should include changes in both the main reaction (CO₂RR) and the competing side reaction (HER). We believe that our manuscript already provides valuable insight into this aspect in a broader context. In addition, while we acknowledge the specialized nature of certain findings, such as the differences in CO₂ configuration under implicit and explicit solvation, we assert the importance of these findings. These observations contribute important insights to the broader field of electrochemical processes at solid-liquid interfaces. To address these concerns and provide clarity on the implications of our work for other studies, we have added an additional paragraph at the end of our manuscript. This paragraph specifically discusses the potential implications and suggestions for future research in related areas that may arise from our findings. We believe that this revision can improve the overall coherence and impact of our work.

Change to manuscript: We have added an additional paragraph to highlight the significance of the molecular understanding and possible inspiration for other studies at the end of Section 2.3 of the manuscript:

"...Typically, determining the reaction network requires a quantitative criterion based on differences in activation barriers. However, in this case, the critical factor is the inhibition of protons approaching the surface by cations. This implies that the concentration of reactants, influenced by potential, ionic activities, and mass transport, may play the dominant role. Furthermore, in reality, the interfacial concentration of alkali metal cations may deviate from one another under a specific potential due to different capacitance responses. Since only a few ions can be introduced at the nanoscale simulations due to scale limitations, it is challenging to reflect the

concentration change accurately. Hence, this necessitates further exploration on both interfacial evolution and simulation methods.

Furthermore, our results derived from the fully explicitly solvated interfacial model offer significant insights into future electrochemical simulation work. Notably, the model indicates different adsorption configurations of the CO₂ molecule in implicit and explicit solvents, distinct coordination structures of solvated alkali metal cations and intermediates, and the interaction between protons and alkali metal cations. The results suggest the importance of explicitly considering the electrolyte in simulation models, particularly when dealing with species that could potentially interact with the interfacial hydrogen bond network or other components within the electrical double layer, including solvated ions. Meanwhile, our investigations into the copper surface may aid in gaining a broader understanding of how alkali metal cations impact CO₂RR performance. This research can be combined with studies conducted on gold surfaces, where the alkali metal cations could differentiate between the CO₂ activation pathways of the inner and outer-spheres. After clarifying the various roles that alkali metal cations play on different coinage metals and even other CO₂RR catalysts, it is expected that a more systematic relationship between electrical double layer design and CO₂RR performance can be established, which demands further exploration."

Reviewer #2:

The authors sent a revised version of the manuscript NCOMMS-23-23847. The authors carefully responded to all the queries and included additional calculations to support their arguments. The authors responded to all the questions I addressed in the initial review of this work, and I consider the manuscript to be suitable for publication in Nature Communications.

Response: We sincerely thank the reviewer for the positive evaluation of our work

I have only a few minor comments that I believe should be addressed beforehand:

1. The title of the manuscript is misleading since the results presented refer to the CO₂ adsorption process on Cu(100) at -0.6 V vs SHE and not CO₂ electroreduction in general.

Response: We thank the reviewer for this valuable suggestion. For clarity, we have revised the title of our manuscript to "*Molecular Understanding of the Critical Role of Alkali Metal Cations in CO₂RR Initiation on Cu(100) Surface*", highlighting the specific reaction step and surface index investigated in the manuscript.

2. The reflection presented in the reply letter regarding the elusiveness of a quantitative criterion in their simulations of the HER should be included in the discussion of the manuscript.

Response: We thank the reviewer for this valuable suggestion. We have extended the corresponding discussions in Section 2.3 of the manuscript:

"...Note that only the qualitative trend is provided here. Typically, determining the reaction network requires a quantitative criterion based on differences in activation barriers. However, in this case, the critical factor is the inhibition of protons approaching the surface by cations. This implies that the concentration of reactants, influenced by potential, ionic activities, and mass transport, may play the dominant role. Furthermore, in reality, the interfacial concentration of alkali metal cations may deviate from one another under a specific potential due to different capacitance responses. Since only a few ions can be introduced at the nanoscale simulations due to scale limitations, it is challenging to reflect the concentration change accurately. Hence, this necessitates further exploration on both interfacial evolution and simulation methods..."

3. This phrase can be a bit confusing "These two aspects result in the abnormal decrease of the proton-interface distance between the proton and the Li⁺ cation at the interface." The authors should try to write it in a clearer way.

Response: We thank the reviewer for this comment. Proton-interface distances are basically affected by two competing processes. One involves the attraction between the proton and negatively charged interface, while the other involves the inhibition of protons from approaching the interface by the introduction of alkali metal cations. For clarity, we have revised the sentence as follows:

"...Thus, the negatively charged catalyst surface attracts protons while the alkali metal cations prevent them from approaching the interface, ultimately leading to distinct proton-interface distances with different alkali metal cations near the interface..."

4. The plots of the total energy of the system as function of time should be included in the SI to show that indeed the systems are equilibrated. The reply letter refers to an incorrect plot and this information was not presented.

Response: We apologize for this mistake and thank the reviewer for this comment. The correct figure has been updated as Supplementary Figure 9 in the Supplementary Information, which is also shown below. The energy drifts are less than 0.03 eV/ps during the sampling period in all systems, indicating their convergence and equilibration.

Supplementary Figure 9. | *Convergence of the systems with proton and various alkali metal cations. Cumulative average of total energy (a) and proton-interface distances (b) of the systems with various alkali metal cations during the sampling period. The equilibration criterion used in our study involves monitoring the energy drift of the systems less than 0.03 eV/ps and the motion of the*

protons less than 0.1 Å/ps for a duration greater than 5 ps to ensure the convergence. The original results of the energies are shown as lighter fluctuating lines in (a), and the original results of (b) correspond to Fig. 5a in the main text. Detailed analysis is shown in Section 2.3 of the main text.

5. Referring to the differences among the cations as size effects should be used with more caution as this could probably be associated with their only difference being their size (the word effect is my main concern is with the word effect in this case). This and the fact that only the effect of alkali metals was analyzed should be clarified.

Response: We thank the reviewer for this comment. We have changed the phrase "size effect" to "*different promotional effects/roles of alkali metal cations ...*". In addition, the word "cations" in some sentences has been clarified as "*alkali metal cations*" to avoid misunderstanding.

REVIEWERS' COMMENTS

Reviewer #2 (Remarks to the Author):

The authors have addressed all the concerns and comments raised in the review. I believe the manuscript is now suitable for publication.